# Sodium Citrate Alleviates Virulence in *Pseudomonas aeruginosa*

**DOI:** 10.3390/microorganisms10051046

**Published:** 2022-05-18

**Authors:** Maan T. Khayat, Tarek S. Ibrahim, Ahdab N. Khayyat, Majed Alharbi, Moataz A. Shaldam, Khadijah A. Mohammad, El-Sayed Khafagy, Dalia A. El-damasy, Wael A. H. Hegazy, Hisham A. Abbas

**Affiliations:** 1Department of Pharmaceutical Chemistry, Faculty of Pharmacy, King Abdulaziz University, Jeddah 21589, Saudi Arabia; tmabrahem@kau.edu.sa (T.S.I.); Ankhayyat@kau.edu.sa (A.N.K.); Maaalharbi1@kau.edu.sa (M.A.); kmohammad@kau.edu.sa (K.A.M.); 2Department of Pharmaceutical Chemistry, Faculty of Pharmacy, Kafrelsheikh University, Kafr El-Sheikh 33511, Egypt; dr_moutaz_986@yahoo.com; 3Department of Pharmaceutics, College of Pharmacy, Prince Sattam Bin Abdulaziz University, Al-kharj 11942, Saudi Arabia; e.khafagy@psau.edu.sa; 4Department of Pharmaceutics and Industrial Pharmacy, Faculty of Pharmacy, Suez Canal University, Ismailia 41552, Egypt; 5Department of Microbiology and Immunology, Faculty of Pharmacy, Egyptian Russian University, Tenth of Ramadan 44629, Egypt; daliadamasy@eru.edu.eg; 6Department of Microbiology and Immunology, Faculty of Pharmacy, Zagazig University, Zagazig 44519, Egypt; hishamabbas2008@gmail.com; 7Pharmacy Program, Department of Pharmaceutical Sciences, Oman College of Health Sciences, Muscat 113, Oman

**Keywords:** *Pseudomonas aeruginosa*, sodium citrate, bacterial virulence, quorum sensing, bacterial resistance, industrial developments

## Abstract

The development of bacterial resistance is an insistent global health care issue, especially in light of the dwindled supply of new antimicrobial agents. This mandates the development of new innovative approaches to overcome the resistance development obstacle. Mitigation of bacterial virulence is an interesting approach that offers multiple advantages. Employing safe chemicals or drugs to mitigate bacterial virulence is an additive advantage. In the current study, the in vitro antivirulence activities of citrate were evaluated. Significantly, sodium citrate inhibited bacterial biofilm formation at sub-MIC concentrations. Furthermore, sodium citrate decreased the production of virulence factors protease and pyocyanin and diminished bacterial motility. Quorum sensing (QS) is the communicative system that bacterial cells utilize to communicate with each other and regulate the virulence of the host cells. In the present study, citrate in silico blocked the *Pseudomonas* QS receptors and downregulated the expression of QS-encoding genes. In conclusion, sodium citrate showed a significant ability to diminish bacterial virulence in vitro and interfered with QS; it could serve as a safe adjuvant to traditional antibiotic treatment for aggressive resistant bacterial infections such as *Pseudomonas aeruginosa* infections.

## 1. Introduction

*Pseudomonas aeruginosa* is a Gram-negative bacterium that causes broadly diverse pathogenesis and illness [1,2]. *P. aeruginosa* causes aggressive infections to almost all body systems; however, it causes serious surgical and burn wound infections, as well as infections of the lung, eye, bloodstream, and urinary tract [3,4]. This splendid capability of *P. aeruginosa* to invade, defeat, and establish infections in host tissues is owed to a huge arsenal of virulence factors. This virulence arsenal expands to involve the production of a wide array of virulent extracellular pigments and enzymes such as protease, elastase, hemolysins, and others for the formation of biofilms, motility, and resistance to oxidative stress [5,6,7]. In a magnificent manner, *P. aeruginosa* employs several systems to orchestrate its virulence factors and regulate its pathogenesis [4]. For instance, *P. aeruginosa* utilizes several types (types 1, 2, 3, 5, and 6) of secretion systems (SS). Although all types of secretion systems are involved in *P. aeruginosa* virulence, T3SS plays an important role in invasion and intracellular survival inside immune cells, as reviewed [8]. Furthermore, the *P. aeruginosa* quorum-sensing (QS) system plays a key role in controlling the production of virulence factors [9]. QS is the chemical language that bacterial cells use to communicate with each other in an inducer–receptor manner [10]. In general, in Gram-negative bacteria, autoinducers of the QS systems are produced by inducer synthetases that bind latterly to surface QS receptors forming inducer–receptor complexes, which have the ability to regulate the expression of virulence factors encoding genes [10,11]. It is well documented that QS controls biofilm formation, bacterial motilities, production of enzymes and pigments, resistance to oxidative stress, and other virulence factors [8,9]. There is growing evidence that targeting QS could guarantee mitigation of bacterial virulence [12,13].

Besides the vigorous virulence of *P. aeruginosa*, it develops phenotypic and/or genotypic resistance to almost all known antimicrobial classes [14]. This gives additional clinical importance to *P. aeruginosa* to be listed among the most important pathogenic microbes [15]. Indeed, resistance development to antibiotics is a major health issue, and the decrease in discovering new antibiotics worsens the situation, resulting in the need to discover new innovative solutions [16,17]. Attenuating bacterial virulence is a reasonable option that confers several advantages. First, mitigating bacterial virulence facilitates their eradication by the immune system and, at the same time, does not affect bacterial growth; hence, it does not induce resistance development [18,19,20]. The maximum benefit is accomplished by employing safe drugs or natural drugs to avoid any probable toxicological effects. In this direction, several drugs, chemical compounds, or natural products were screened for their antivirulence activities [12,21,22,23,24,25,26].

Sodium citrate is commonly used as an emulsifier for oils and is used in food industries as an acidity regulator and sanitizer by lowering the pH, providing unsuitable conditions for bacterial growth. It is also used in the collection of blood samples to prevent clotting in storage [27,28]. Furthermore, sodium citrate is used to neutralize excess acid in the urine and blood, as well as in the treatment of chronic kidney diseases and metabolic acidosis [28]. Importantly, it was shown that sodium citrate has antimicrobial activity, independent of pH, against oral *Streptococcus pneumoniae* and several oral bacteria such as *Fusobacterium nucleatum* and *Streptococcus mutans* [27]. Importantly, it was reported that sodium citrate (4%) could inhibit the formation of *Klebsiella pneumoniae* biofilms by 46.5% [29]. In another study, 4% sodium citrate in solution with 0.0015% nitroglycerin and 22% ethanol could eradicate biofilm formed by methicillin-resistant *Staphylococcus aureus* (MRSA), methicillin-resistant *Staphylococcus epidermidis* (MRSE), vancomycin-resistant enterococci (VRE), multidrug-resistant *Klebsiella pneumoniae*, *P. aeruginosa*, *Acinetobacter baumannii*, Enterobacter cloacae, *Escherichia coli*, and *Stenotrophomonas maltophilia*, in addition to *Candida albicans* and *Candida glabrata*. Furthermore, sodium citrate at a concentration of 4% was able to prevent the biofilm formation of *K. pneumoniae*, *Staphylococcus aureus*, and *Escherichia coli* [30,31,32,33]. In this context, the antibiofilm and antivirulence activities of sodium citrate were evaluated at concentrations of 4% and 5% against *P. aeruginosa*.

Taking into consideration that sodium citrate has no known toxicological reaction [28], this study aimed to evaluate the antivirulence activities of sodium citrate. In the current study, the antivirulence and anti-QS activities of sodium citrate against *P. aeruginosa* were investigated in vitro and in silico.

## 2. Materials and Methods

### 2.1. Chemicals and Bacterial Strains

All microbiological media were purchased from Oxoid (Hampshire, UK). All the used chemicals and sodium citrate were of pharmaceutical grade and purchased from Sigma-Aldrich (St. Louis, MO, USA). *P. aeruginosa* PAO1 was used in this study

### 2.2. Determination of Sodium Citrate Effect on Bacterial Growth

In order to ensure the antivirulence effect of sodium citrate is not due to the inhibition of bacterial growth, the effect of sodium citrate at tested concentrations on *P. aeruginosa* growth was assessed, as described previously [34]. Briefly, fresh *P. aeruginosa* cultures were inoculated overnight in LB broth provided with 4% or 5% sodium citrate at 37 °C for 24 h. The turbidites of *P. aeruginosa* cultures were measured at 600 nm, and viable bacterial cells were counted.

### 2.3. Evaluation of Antibiofilm Activities of Sodium Citrate

The inhibition of *P. aeruginosa* biofilm formation by sodium citrate was assessed by the crystal violet method [35]. One hundred microliter aliquots of *P. aeruginosa* suspension of an approximate cell inoculum of 1 × 106 CFU/mL were transferred to microtiter plate wells in the presence or absence of sodium citrate (4% and 5%). The nonadherent cells were washed out after 24 h incubation at 37 °C, and the biofilm-forming cells were fixed with methanol and stained with crystal violet (1%) for 20 min. The excess dye was washed out, plates were air-dried, adhered dye was extracted with 33% glacial acetic acid, and absorbances were measured at 590 nm using the Biotek Spectrofluorimeter (Winooski, VT, USA).

### 2.4. Assessment of Sodium Citrate Effect on P. aeruginosa Motility

The sodium citrate inhibition of *P. aeruginosa* swarming motility was performed as described previously [17,36]. LB agar plates containing 4% or 5% sodium citrate and control LB agar plates without sodium citrate were centrally inoculated with 5 µL of fresh *P. aeruginosa* PAO1 culture prepared from an overnight culture in tryptone broth, and the swarming zone was measured.

### 2.5. Determination of Sodium Citrate Effect on Pyocyanin Production

The virulent *P. aeruginosa* pyocyanin pigment was assayed in the presence or absence of sodium citrate, as previously shown [37,38]. Ten microliter aliquots of *P. aeruginosa* overnight cultures (adjusted to OD600 of 0.4) were mixed with 1 mL of LB broth provided with sodium citrate (4% or 5%). After 48 h incubation at 37 °C, the Eppendorf tubes were centrifuged, and the absorbances of pyocyanin pigment in the supernatants were measured at 691 nm.

### 2.6. Evaluation of Inhibitory Effect on Protease Activity

The skim milk agar method was used to assess the inhibitory effect of sodium citrate on the activity of protease [39]. *P. aeruginosa* overnight cultures in the presence or absence of sodium citrate (4% or 5%) were centrifuged to obtain the extracellular protease in the supernatants. One hundred microliters of the supernatants was placed in the wells prepared in skim milk (5%) agar plates. After 24 h incubation at 37 °C, the clear zones representing the proteolytic activity were measured.

### 2.7. Assessment of Sodium Citrate Effect on QS-Encoding Genes

A quantitative real-time PCR was performed to attest to the effect of sodium citrate on the expression of QS-encoding genes in *P. aeruginosa*. Citrate-treated and untreated overnight cultures of PAO1 at 37 °C were prepared, and the pellets were collected by centrifugation at 12,000× *g* for 2 min. The pellets were resuspended in Tris–EDTA buffer with lysozyme (100 µL) and incubated for 5 min at 25 °C. The lysis buffer with β-mercaptoethanol was added and mixed well. The RNA of *P. aeruginosa* cultures treated or not with sodium citrate 5% was extracted (Purification Kit Gene JET RNA, Thermo Scientific, Waltham, MA, USA) and stored at −80 °C as described [40]. The expression levels were normalized to the housekeeping gene *ropD*, and the primers are listed in Table 1. The cDNA was synthesized using a high-capacity cDNA reverse transcriptase kit (Applied Biosystem, Waltham, MA, USA) and amplified using the SYBR Green I PCR Master Kit (Fermentas, Waltham, MA, USA) in a Step One instrument (Applied Biosystem, Waltham, MA, USA). A melting curve was established according to the manufacturer, and the relative expressions were calculated using the comparative threshold cycle (∆∆Ct) method [41].

### 2.8. In Silico Assessment of Sodium Citrate Ability to Bind P. aeruginosa QS Receptors

The *P. aeruginosa* LasR receptor (PDB ID: 2UV0) [42], RhlR receptor model (ID: P54292) [7], and citrate [43] were downloaded, then prepared using AutoDockTools [44] in accordance with our prior procedures [7]. AutoDock Vina [45] was used for docking, while Discovery studio [46] was used for both 3D visualization and 2D schematic presentation.

### 2.9. Statistical Analysis

All the performed experiments were conducted in triplicates, and the data are expressed as means ± standard errors. One-way ANOVA test, followed by Tukey’s post-test, was used (unless mentioned) to test the statistical significance, where *p* ≤ 0.05 was considered significant.

## 3. Results

### 3.1. Sodium Citrate at Concentrations of 4% or 5% Does Not Affect P. aeruginosa Growth

To exclude the effect of sodium citrate on bacterial growth, the optical densities of *P. aeruginosa* growth were measured in the presence or absence of 4% or 5% sodium citrate. There was no significant difference between bacterial growth in the presence or absence of sodium citrate (Figure 1). The bacterial cell count was performed, and there were no significant differences between counts of *P. aeruginosa* cultures treated or not with sodium citrate (Appendix A).

### 3.2. Sodium Citrate Inhibits P. aeruginosa Biofilm Formation

To assess the sodium citrate antibiofilm effect, the absorbances of stained biofilm-forming cells with crystal violet were measured in the presence or absence of 4% or 5% sodium citrate. The results were expressed as percentage change from untreated *P. aeruginosa* control. Sodium citrate in concentrations of 4% or 5% significantly decreased biofilm formation by 74.64% and 76.02%, respectively (Figure 2).

### 3.3. Sodium citrate Diminishes P. aeruginosa Motility

*P. aeruginosa* motility ensures its spread in the host tissues and enhances its pathogenicity [7]. The diameters of swarming motility of *P. aeruginosa* on agar plates provided with 4% or 5% sodium citrate were measured. Sodium citrate at concentrations of 4% or 5% significantly diminished bacterial motility by percentages of 80% and 87.6%, respectively (Figure 3).

### 3.4. Sodium Citrate Decreases the P. aeruginosa Pigment Pyocyanin

Pyocyanin, the bluish-green pigment produced by *P. aeruginosa*, has emerged as an important virulence factor that aids in killing host cells, as well as competitor microbes [47]. The absorbance of the produced pyocyanin was measured in *P. aeruginosa* treated or not with sodium citrate (4% or 5%). The data are presented as percentage change from untreated control. Sodium citrate significantly reduced pyocyanin production by 78.5% and 81.5% for concentrations of 4% and 5%, respectively (Figure 4).

### 3.5. Sodium Citrate Decreases the Production of Protease

*P. aeruginosa* produces a wide array of extracellular virulent enzymes to establish and spread its infection into the host tissues. Protease facilitates the spread of bacterial infection, and the decrease in its production mitigates bacterial virulence [2]. The sim milk agar method was used to assess the effect of sodium citrate on protease activity. The extracellular protease collected from *P. aeruginosa* cultures treated with or without 4% or 5% sodium citrate was poured in wells made in skim milk agar plates, and the clear zones were measured. Sodium citrate significantly decreased the production of protease by 58.7% at a concentration of 4%, and protease inhibition was 100% at a concentration of 5% (Figure 5).

### 3.6. Sodium Citrate Anti-QS Activities

#### 3.6.1. Sodium Citrate Downregulates the *P. aeruginosa* QS Genes

*P. aeruginosa* mainly utilizes three QS systems to regulate the production of its virulence factors [48]. The expression of the encoding genes of the autoinducer synthetases and their receptors in the three QS systems were quantified using RT-PCR in the presence or absence of 5% sodium citrate. The experiment was repeated in triplicate, the fold change of expression levels was represented as mean ± SD, and Student’s t-test was employed to attest to the significance. The current data revealed a significant reduction in the expression of all the QS-encoding genes in the presence of sodium citrate (Figure 6).

#### 3.6.2. Sodium Citrate Interferes with the Binding of Autoinducers to *P. aeruginosa* QS Receptors

The quorum-sensing proteins LasR and RhlR were identified as potential targets for *P. aeruginosa* virulence inhibition [49]. A molecular docking investigation was performed to reveal the binding mechanism for citrate as a potential inhibitor for LasR and RhlR. Molecular docking demonstrated the good binding affinity of citrate for LasR (affinity = −5.9 Kcal/mol) that was comparable to the natural ligand (affinity = −6.1 Kcal/mol). Additionally, citrate had a respectable binding affinity to RhlR (affinity = −5.3 Kcal/mol) when related to C4-HSL (affinity = −5.8 Kcal/mol). The key interactions of citrate with LasR and RhlR are presented in Figure 7 and Figure 8, respectively.

Citrate had attractive charges with Arg61 and Arg112 for LasR and RhlR, respectively. Inside the LasR active site, citrate had a pi-anion interaction with Trp88. In addition, H-bonding with Tyr64, Thr115, and Ser129 was observed with the oxygen of the carboxylate groups of citrate into LasR, while only Trp108 formed an H-bond with citrate inside the active site of RhlR. All these mentioned interactions, besides the hydrophobic interactions, shown in Figure 7 and Figure 8 contributed to the affinity of citrate to LasR and RhlR targets as a potential quorum-sensing inhibitor.

## 4. Discussion

In this work, the antivirulence activity of sodium citrate against *P. aeruginosa* was assessed. *P. aeruginosa* is an excellent bacterial model to understand bacterial virulence, not only because of its arsenal of virulence factors but also its remarkable ability to develop resistance to different classes of antibiotics [2,4,15,50,51]. Antivirulence therapy is based on using a safe adjuvant to mitigate bacterial virulence without affecting bacterial growth. This approach is much less likely to lead to the emergence of resistance. Moreover, it enhances the immune system to eradicate microbial infection and augment antibiotic activity [9,17,52].

To exclude the probability that the antivirulence activity of sodium citrate is due to the inhibition of bacterial growth, the effect of sodium citrate at selected concentrations on *P. aeruginosa* growth was investigated, and sodium citrate did not interfere with bacterial growth. This means all subsequent antivirulence activities are not due to the inhibition of bacterial growth and are apart from the acidification influence of sodium citrate on the surrounding medium. The bacterial biofilms are additional obstacles to efficient antibiotic treatment, as observed in chronic and nosocomial infections, and so the biofilm eradication is a golden target in such infections [53,54]. The present findings showed the significant ability of sodium citrate to inhibit the biofilms formed by *P. aeruginosa* to more than 70% at concentrations of both 4% and 5%. Bacterial motility is an important structural virulence factor that eases the spread of bacterial infection and is associated with biofilm formation [20,29,54,55]. *P. aeruginosa* is peritrichous and can swim, swarm, and slide on solid surfaces [7,56]. Sodium citrate significantly prevented *P. aeruginosa* swarming at both selected concentrations.

*P. aeruginosa* pathogenesis is accomplished by employing diverse enzymes such as proteases, lipases, hemolysins, elastase, and others [4,17]. Protease aids bacteria to destroy the host tissue, conferring great ability for infection spread and conquering host defense [25,57]. Sodium citrate at 4% significantly diminished the activity of protease by *P. aeruginosa*; however, this inhibition was complete at a concentration of 5%. Besides enzymes, *P. aeruginosa* produces its characteristic bluish-green pigment pyocyanin, the roles of which in the virulence and survival of *P. aeruginosa* are well documented [37,47,58]. Our findings showed that sodium citrate at 4% or 5% could significantly reduce the production of pyocyanin.

Bacterial QS is used by bacterial cells to orchestrate the expression of virulence factors during the course of infection [5]. Both Gram-negative and -positive bacterial cells depend on QS systems to orchestrate the expression of virulence factors during the course of infection [24,59]. In Gram-negative, a wide array of autoinducers is produced and released to the surrounding niche, where they bind to their specific cognate receptors [26,60]. Then, the autoinducer–receptor complex binds to a specific DNA sequence to regulate the expression of virulence genes [34,60]. For instance, QS receptors LuxR, which are widely detected in different Gram-negative genera, bind to autoinducers to form complexes that bind to short DNA sequences on the bacterial chromosome called lux boxes to regulate their downstream virulent genes [61]. The QS system controls diverse *P. aeruginosa* virulence factors, including biofilm formation, motility and production of pyocyanin, and extracellular enzymes such as protease, as extensively documented [60,62,63,64]. The anti-QS activity of sodium citrate was assessed genotypically, and it significantly downregulated *P. aeruginosa* QS genes. Citrate downregulated the expression of the main three QS-receptor-encoding genes in *P. aeruginosa lasR*, *rhlR*, and *pqsR*, in addition to decreasing the production of inducer-synthetase-encoding genes *lasI*. *rhlI*, and *pqsA*. Furthermore, and in agreement with the above results, sodium citrate binds to *P. aeruginosa* QS receptors, competing with autoinducers in the in silico study.

These findings declare the antivirulence activities of sodium citrate at concentrations of 4% or 5%. The antivirulence and antibiofilm activities of sodium citrate could be owed to interference with QS systems, as sodium citrate binds to QS receptors and downregulates the expression of QS-encoding genes. As declared previously, sodium citrate was used efficiently alone or in combinations to eradicate the biofilms formed by different bacterial strains on inanimate objects or on living tissues [27]. The safety plus ability to diminish bacterial virulence at low concentrations indicates the possible application of sodium citrate as an adjuvant to traditional antibiotics in the treatment of aggressive bacterial infections caused by *P. aeruginosa.*

## 5. Conclusions

Conquering bacterial resistance requires looking for new approaches such as using efficient adjuvants to traditional antibiotics. These agents must possess some criteria, be safe, and not affect bacterial growth to avoid bacterial resistance development. This study evaluated the antivirulence activities of sodium citrate against *P. aeruginosa*. The present data showed the significant in vitro ability of sodium citrate to mitigate bacterial virulence, inhibiting biofilm formation and motility and reducing the production of *P. aeruginosa pyocyanin* pigment and the activity of the protease enzyme. The antivirulence activities of citrate were attributed to its ability to interfere with QS systems. This study proclaims the possible application of sodium citrate as an antivirulence agent and as an adjuvant to antibiotics. However, future work is needed to confirm the antivirulence activity in animal models.

## Figures and Tables

**Figure 1 microorganisms-10-01046-f001:**
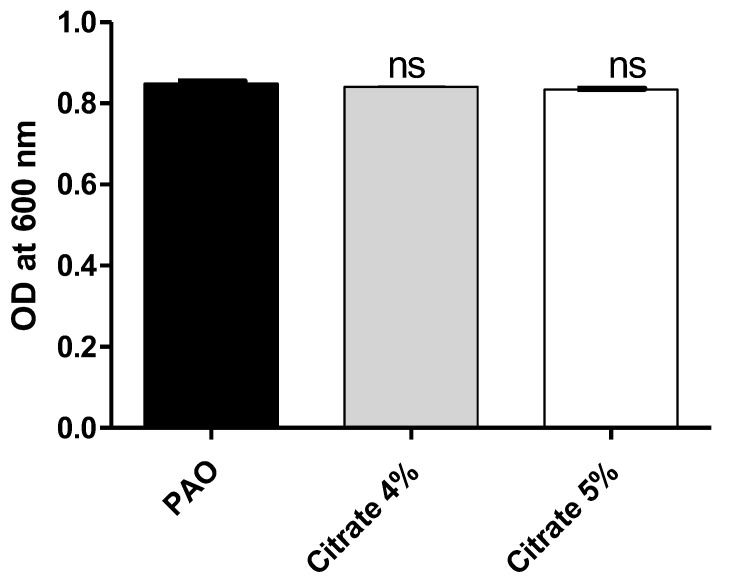
Sodium citrate does not affect *P. aeruginosa* growth. The optical densities of *P. aeruginosa* growth were measured at OD_600_ after 24 h incubation in the presence and absence of 4% or 5% sodium citrate. ns: non-significant.

**Figure 2 microorganisms-10-01046-f002:**
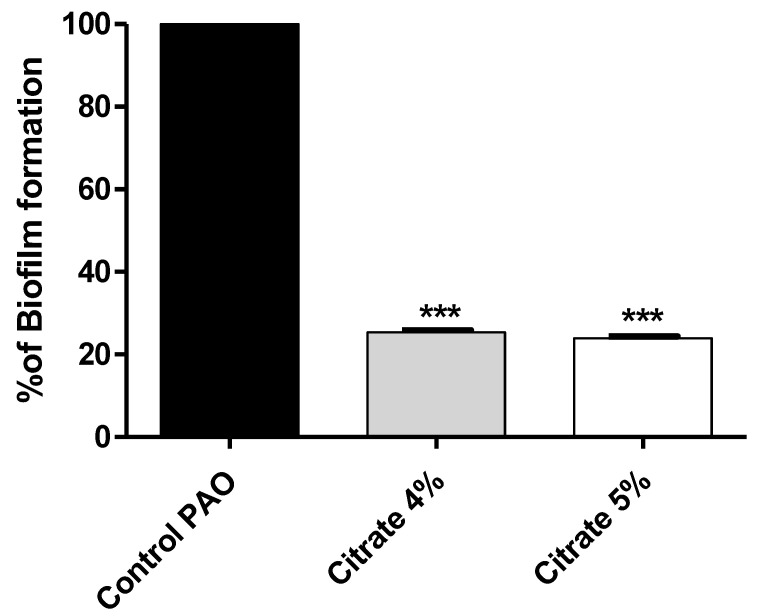
Sodium citrate inhibits biofilm formation in *P. aeruginosa*. The absorbances of crystal-violet-stained biofilm-forming cells were measured. Sodium citrate in 4% or 5% significantly decreased biofilm formation (*** = *p* < 0.0001).

**Figure 3 microorganisms-10-01046-f003:**
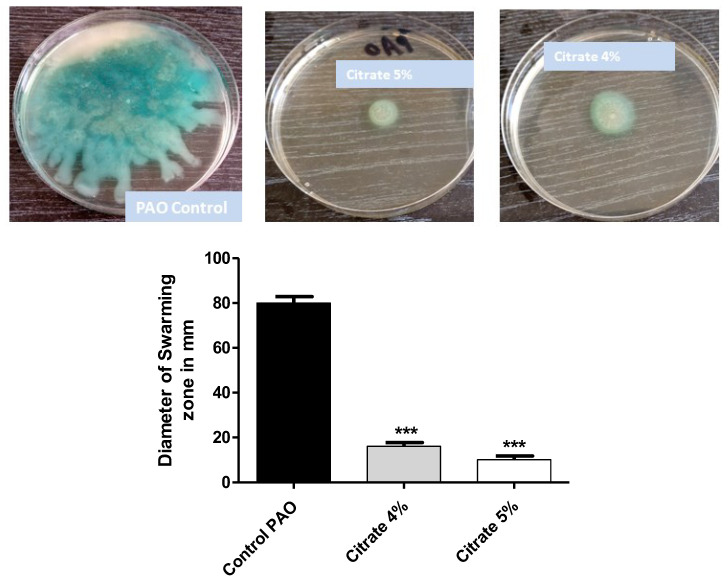
Sodium citrate curtails *P. aeruginosa* motility. The diameters of *P. aeruginosa* swarming were measured in agar plates provided or not with 4% or 5% sodium citrate. Sodium citrate significantly diminished bacterial motility (*** = *p* < 0.0001).

**Figure 4 microorganisms-10-01046-f004:**
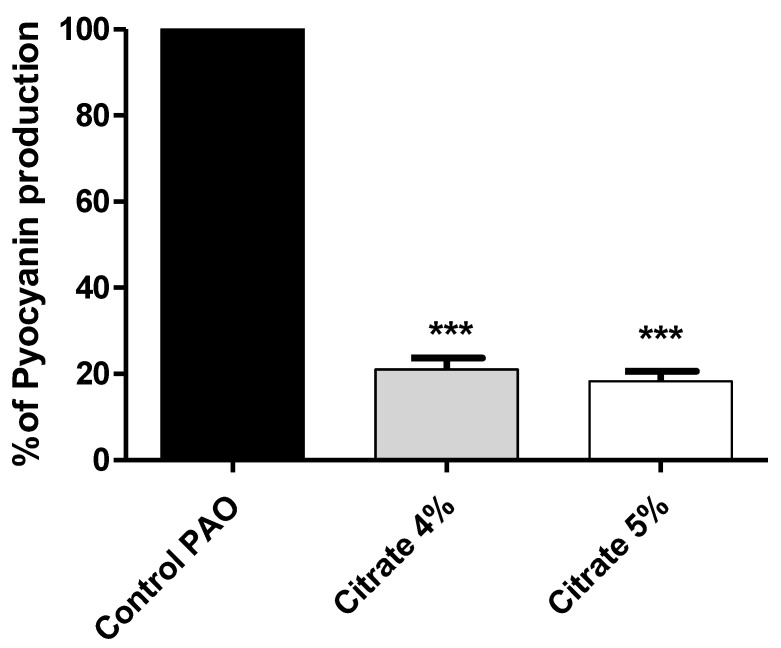
Sodium citrate reduces the production of *P. aeruginosa* virulent pigment. The absorbances of produced pyocyanin were measured in cultures provided or not with 4% or 5% sodium citrate. Sodium citrate significantly reduced pyocyanin production (*** = *p* < 0.0001).

**Figure 5 microorganisms-10-01046-f005:**
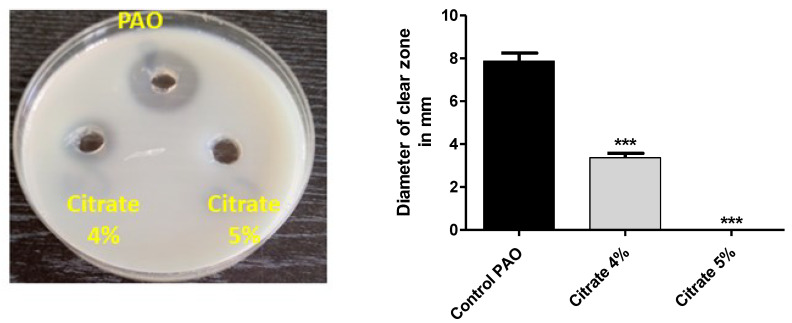
Sodium citrate decreases the activity of protease. The clear zones due to proteolytic effects of collected protease from cultures provided or not with 4% or 5% sodium citrate on skim milk agar were measured. Sodium citrate significantly reduced protease production (*** = *p* < 0.0001).

**Figure 6 microorganisms-10-01046-f006:**
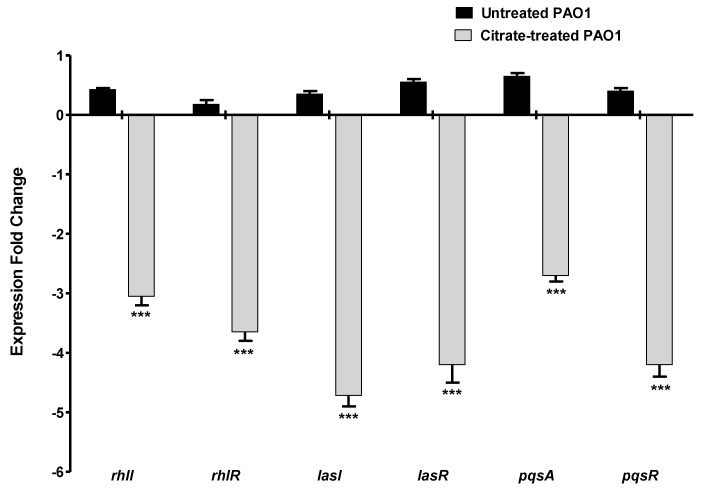
Sodium citrate downregulates the expression of QS-encoding genes. The expressions of the encoding genes of the autoinducer synthetase and receptors of the main three QS systems in *P. aeruginosa* were quantified using RT−PCR and normalized to the expression level of housekeeping gene *ropD*. Sodium citrate significantly decreased the expression of all QS-encoding genes (*** = *p* < 0.0001).

**Figure 7 microorganisms-10-01046-f007:**
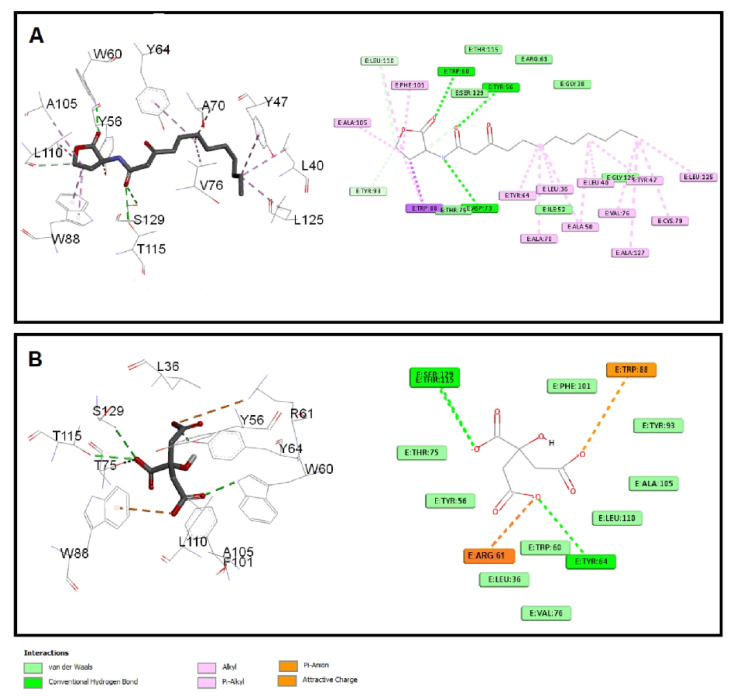
Molecular docking of (**A**) citrate and (**B**) C12-HSL into the active site of LasR protein 3D representation (left) and 2D schematic interaction (right). Citrate could bind with the LasR receptor and interfere with the *P. aeruginosa* QS systems.

**Figure 8 microorganisms-10-01046-f008:**
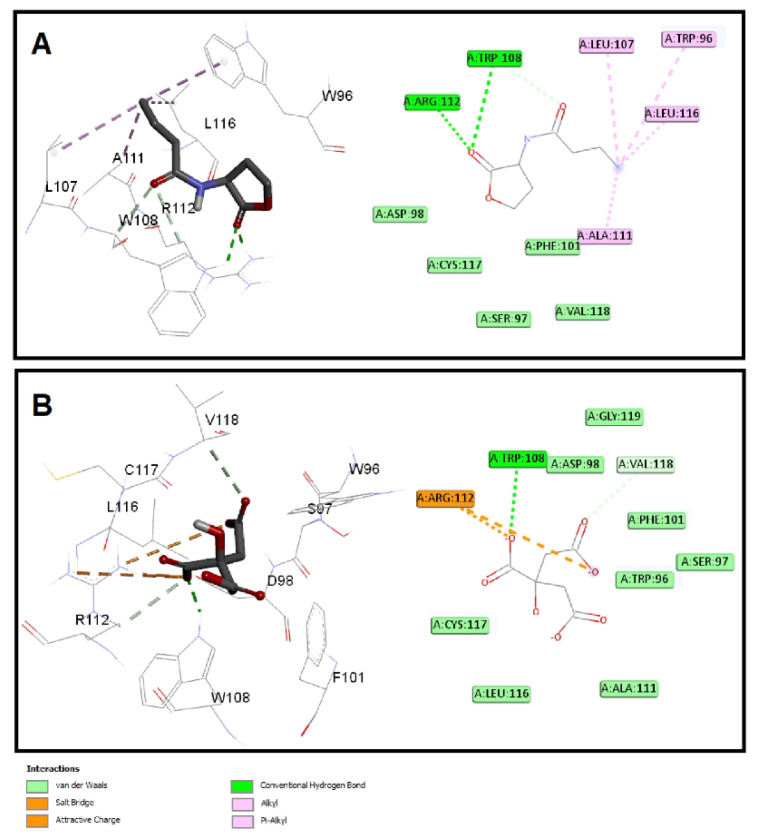
Molecular docking of (**A**) citrate and (**B**) C4-HSL into the active site of RhlR protein 3D representation (left) and 2D Schematic interaction (right). Citrate could bind with the RhlR receptor and interfere with the *P. aeruginosa* QS systems.

**Table 1 microorganisms-10-01046-t001:** Sequences of the used primers in this study [17].

Target Gene	Sequence (5′–3′)
*lasI*	**For:** CTACAGCCTGCAGAACGACA **Rev:** ATCTGGGTCTTGGCATTGAG
*lasR*	**For:** ACGCTCAAGTGGAAAATTGG **Rev:** GTAGATGGACGGTTCCCAGA
*rhlI*	**For:** CTCTCTGAATCGCTGGAAGG **Rev:** GACGTCCTTGAGCAGGTAGG
*rhlR*	**For:** AGGAATGACGGAGGCTTTTT **Rev:** CCCGTAGTTCTGCATCTGGT
*pqsA*	**For:** TTCTGTTCCGCCTCGATTTC **Rev:** AGTCGTTCAACGCCAGCAC
*pqsR*	**For:** AACCTGGAAATCGACCTGTG **Rev:** TGAAATCGTCGAGCAGTACG
*rpoD*	**For:** GGGCGAAGAAGGAAATGGTC **Rev:** CAGGTGGCGTAGGTGGAGAAC

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
