# Peer review of "Sodium Citrate Alleviates Virulence in Pseudomonas aeruginosa"

_microorganisms, 2022, doi:10.3390/microorganisms10051046_

Round 1
Reviewer 1 Report
The work of Maan T. Khayat and co-workers describes the anti-virulence and anti-QS activities of sodium citrate against P. aeruginosa evaluated in-vitro and in-silico.
I suggest that the authors consider the following points in preparing the final manuscript:
- The manuscript must be improved because there are some confuse sentences like this: " Pseudomonas aeruginosa is an abundant Gram-negative bacterium and causes wide diverse pathogenesis and illness.”
- What apparatus did the authors use to measure the absorbance?
- By what method did the authors count the bacterial cells?
- Please check the bacteria names (italics).
Author Response
Dear Reviewer,
We appreciate the reviewer's valuable and constructive comments and suggestions, which greatly helped us to improve the manuscript. Please find the attached response to the points you raised.
Thank you,
Best Regards,
Wael

Reviewer 2 Report
General Considerations: In the manuscript entitled “Sodium citrate alleviates virulence in Pseudomonas aeruginosa” the effect of 4% and 5% of sodium citrate was evaluated on the P. aeruginosa biofilms formation, motility, pyocyanin production, protease production and on the expression of quorum sensing encoding genes. The results show that sodium citrate affects some of the virulence factors of P. aeruginosa, however the effect of this compound on the virulence of these bacteria was not evaluated using an infection model. Generally, the manuscript is objective and clearly presented, but some of the inferences and conclusions drawn, in my opinion, need to be revised.
Minor comments:
- Line 49-50: “However, all types of secretion systems participate in the aeruginosa virulence” could be replaced by “Although, all types of secretion systems are involved in P. aeruginosa virulence”
- Line 51-52: “Furthermore, aeruginosa quorum sensing (QS) system plays the key role in regulation of the production virulence factors” could be replaced by “Furthermore, P. aeruginosa quorum sensing (QS) system plays a key role in control the production of virulence factors”.
- Line 54: “with each other in and inducer-receptor manner” should be replaced by “with each other in an inducer-receptor manner”.
- Line 54-56: “In general, QS systems in Gram-negative bacteria, autoinducers are produced by inducer synthetases that bind latterly to surface QS receptors forming induce-receptor complexes which has the ability to regulate the expression of virulence factors encoding genes” could be replaced by “In general, in Gram negative bacteria, autoinducers of the QS systems are produced by inducer synthetases that bind latterly to surface QS receptors forming induce-receptor complexes, which have the ability to regulate the expression of virulence factors encoding genes”
- Line 62: Instead of “typic resistance to almost known antimicrobial classes” it should be “typic resistance to almost all known antimicrobial classes”
- Line 82-84: The sentence is not complete, a verb is missing.
- Line 159, 169: aeruginosa should be italicized.
- Line 164: Instead of “crysatl” it should be “crystal” and instead of “measuere” it should be “measured”.
- Line 185: “Pyocyanin the bluish green pigment produced by aeruginosa has emerged” should be replaced by “Pyocyanin, the bluish green pigment produced by P. aeruginosa, has emerged”.
- Line 215: “Rt-PCR” should be replaced by “RT-PCR”.
- Line 233, 250, 254 and 255: “rhlr” should be replaced by “RhlR”.
- Line 251: A full stop is missing in the end of the sentence: “Inside LasR active site, citrate had a p-anion interaction with Trp88.”
Major comments:
- To test the ability of aeruginosa to produce biofilms it is described in the Materials and Methods section that “One hundred µL aliquots of P. aeruginosa (OD600 of 0.4) were transferred to of microtiter plates wells in the presence or absence of sodium citrate 100 4% or 5%.”. This initial OD value is much higher than the amount generally mentioned in the bibliography. If it was not a mistake in writing, could you please justify this option?
- Could you please mention the growing conditions used and the time point/OD selected to collect the samples for RNA extraction and the RT-PCR?
- In the Results section, it is mentioned that “there was no significant difference between the bacterial growth in the presence or absence of sodium citrate”, however only a time point (24h after incubation, late stationary phase) is presented in Figure 1. Has a growth curve been made over time? How can you guarantee that there are no growth differences during the exponential phase?
- Could you please show the raw data of biofilms, RT-PCR and pyiocyanin production in a supplementary file?
- Although the effect of sodium citrate in several P. aeruginosa virulence factors have been tested, no infection model was used to evaluate the effect of this compound in P. aeruginosa virulence. However, in the title of the manuscript, in the abstract, results and discussion sections it is mentioned several times that sodium citrate alleviates/ show a significant ability to diminish the virulence of aeruginosa. I strongly recommend that these inferences should be reviewed.
Author Response

(The authors gave the same response as above.)

Reviewer 3 Report
In this article, the authors describe the use of sodium citrate as an antimicrobial agent to control Pseudomonas aeruginosa infections. The authors claim that sodium citrate can diminish virulence and interfere with the quorum sensing system in P. aeruginosa, making it a potential candidate to control P. aeruginosa virulence. Though the study is novel and promising, the authors need to revise the manuscript considering the suggestions given below before it can go for publication. The manuscript needs to be modified giving emphasis to language usage, sentence structure, grammar and the usage of italics.
Major comments
- Introduction: Last paragraph à please elaborate the part ‘antimicrobial activity of sodium citrate in other bacterial strains including oral bacteria (name it)’. There are other published works showing the antimicrobial activity of sodium citrate in other bacterial strains. Please describe those as well.
- The authors show that the expression of QS receptors are decreased in the presence of sodium citrate. It is better to explain in the discussion session how the phenotypic properties like biofilm formation, motility and the secretion of protease are affected by the decreased expression of QS genes.
- Need to avoid un-appropriate self-citation
Minor comments
L54: an inducer-dependent
L60: grantee?
L62: This gives additional
L91: not due to the inhibition of
L129: rpoD
L200: with or without
L212: down-regulates P. aeruginosa QS genes
L216: in triplicates
L250: RhlR
L255: Figure 7 and 8
The discussion session needs to be more concise.
First paragraph of Discussion: It does not need to be elaborate like introduction.
Second paragraph: The details of the antimicrobial effect and the uses of sodium citrate should go to the ‘introduction’ session.
Fifth paragraph: No need to give long introduction about QS in Gram –ve bacteria. L308 to L318: Concise it to 2-3 sentences
L319, 320: QS genes
Overall, the manuscript needs to be written more concisely.
Citations: Please provide only the most relevant ones and remove the irrelevant references. If there are multiple references for the same idea, provide only the first published one. There are a lot of citations which are quite newly published though the idea was reported by other researchers long back. Also, provide only one reference (the first paper published) for each target gene in Table 1.
Line 39: Citation 1-2; Pseudomonas aeruginosa is an abundant Gram-negative bacterium and causes wide diverse pathogenesis and illness. Citations are quite new and the pathogenesis of P. aeruginosa is reported long back. Need to avoid un-appropriate self-citation.
Line 57-58: It is well documented that QS controls the biofilm formation, bacterial motilities, production of enzymes and pigments, resistance to oxidative stress, and other virulence factors [7,8,17]. All the references are published in 2021. The authors should provide the original references for the above idea instead of self-citation.
Line 59 – 60: There is growing evidence that targeting QS could grantee mitigation of bacterial virulence [16,18-20]. All the citations are from 2021 (self-citation) though there are a lot of papers which are already published earlier regarding the same.
Line 53: QS is the chemical language which bacterial cells use to communicate with each other in and inducer-receptor manner [10-15]. This is known for decades. The citations 10-15 are newer ones (year 2020-2022) and belongs to self-citation.
Line 57: Reference 16 à Self citation, newly published.
Author Response

(The authors gave the same response as above.)

Round 2
Reviewer 2 Report
Dear authors,
Thank you for your comments and explanations. I consider that your changes have improved the manuscript.
Please review the following details:
Line 83: Staphylococcus aureus should be italicized.
Line 242: “Sodium citartes” should be replaced by “ Sodium citrate”.
Line 325: "Qs-" should be replaced by "QS".
Author Response
Dear Reviewer,
Thank you for your comments and explanations. I consider that your changes have improved the manuscript.
Please find attached corrections.
Best Regards,
Wael

Reviewer 3 Report
Nil
Author Response
Dear Reviewer,
We appreciate the reviewer's valuable and constructive comments and suggestions, which greatly helped us to improve the manuscript.
Best Regards,
Wael